# Safety of Anti-Angiogenic Drugs in Pediatric Patients with Solid Tumors: A Systematic Review and Meta-Analysis

**DOI:** 10.3390/cancers14215315

**Published:** 2022-10-28

**Authors:** Andrea Spini, Valerio Ciccone, Pietro Rosellini, Marina Ziche, Ersilia Lucenteforte, Francesco Salvo, Sandra Donnini

**Affiliations:** 1Department of Medicine, Surgery and Neuroscience, University of Siena, 53100 Siena, Italy; 2Azienda Ospedaliera Universitaria Senese, 53100 Siena, Italy; 3Department of Life Sciences, University of Siena, 53100 Siena, Italy; 4CIC 1401, CIC Bordeaux, 33000 Bordeaux, France; 5Department of Clinical and Experimental Medicine, University of Pisa, 56126 Pisa, Italy; 6INSERM, BPH, U1219, Team Pharmacoepidemiology, Université de Bordeaux, 33000 Bordeaux, France

**Keywords:** anti-VEGF, vegf inhibitors, antiangiogenic drugs, pediatric, children, childhood, cancer, tumor, meta-analysis, systematic review

## Abstract

**Simple Summary:**

Given the low prevalence and the heterogeneity of childhood cancers, information about the safety of anti-angiogenic drugs in pediatric patients is only partially assessed. We aimed to evaluate the safety of these drugs in children with solid tumors. This systematic review and meta-analysis reported that one out of two pediatric patients using anti-angiogenic drugs in monotherapy experienced a serious adverse event despite proportions varying per single drug.

**Abstract:**

Cancer is a clinical condition that can benefit from anti-angiogenic drugs (AADs). Given the low prevalence and the heterogeneity of childhood cancers, information about the safety of these drugs in pediatric patients is partially assessed. The aim of this study was to evaluate the safety of AADs in pediatric patients with solid tumors. Clinical trials and observational studies were searched in PubMed, ISI Web of Science, and ClinicalTrials database For each included study, adverse events (AEs) were extracted. A meta-analysis was conducted by pooling proportions of AEs using a random intercept logistic regression model. Seventy studies were retrieved. Most part were clinical trials (55 out of 70), and only fifteen observational studies were found. Overall, proportion of serious and non-serious AEs of AADs used as monotherapy was 46% and 89%, respectively. Proportions of serious AEs varied among drugs: sunitinib, 79%; lenvatinib, 64%; sorafenib, 48%; ramucirumab, 41%; pazopanib, 30%; and vandetanib, 27%. A higher proportion of non-serious hematological AEs was found in the patients receiving pazopanib with respect to sunitinib and lenvatinib. The safety profile of AADs has been extensively investigated for mostly drugs in phase I and II trials and is limited to acute toxicities. Overall, one out of two patients using AAD drugs in monotherapy experienced a serious AE despite proportions varied per single drugs. When AADs were combined with standard chemotherapy, the proportion of AEs varied in relation to the single combinations.

## 1. Introduction

Aberrant tumor vessels in solid tumors contribute to maintaining the pro-tumorigenic niche and profoundly influence the success of anticancer therapies [1]. Tumor angiogenesis is mainly driven by an imbalance between pro-angiogenic and anti-angiogenic signaling in the tumor microenvironment. A key pro-angiogenic mediator is vascular endothelial growth factor (VEGF), but other factors can stimulate angiogenesis including fibroblast growth factor-2 (FGF-2); platelet-derived growth factor (PDGF); hepatocyte growth factor (HGF); angiopoietins, and inflammatory mediators, such as interleukins and prostaglandins [2]. VEGF is crucial for tumor angiogenesis, and most antiangiogenic drugs are directed against this factor or its receptors, such as bevacizumab and ramucirumab [3]. However, to avoid resistance to anti-VEGF drugs, over recent decades, alternative strategies that simultaneously target VEGF signaling pathway and other pro-angiogenic signals, including tyrosine kinase inhibitors (TKIs) such as sunitinib, sorafenib, pazopanib, cabozantinib, and others were developed (Table 1) [4,5].

Despite advances in anticancer therapies, pediatric malignancies continue to be a leading cause of death by disease in people younger than 20 years of age, and while in recent decades, the survival has improved for leukemia and lymphomas, it reached a plateau for many solid tumors [6]. Solid tumors account for 30% of all pediatric cancers. In children, the most common solid tumors are neuroblastoma, central nervous system tumors, sarcomas and Wilms’ tumor [7]. Recent advancements in understanding pediatric tumors have arisen from evaluation of the complex genetic landscape within each tumor subtype. Although for anti-angiogenic drugs there are no clear predictive biomarkers of response, these drugs are considered promising chemosensitizers of anticancer strategies such as chemotherapy, targeted therapies, and immune therapies in several advanced tumors, and while not approved, they are frequently used in pediatric population [5,8,9]. For example, benefits of anti-angiogenic therapy in brain tumors are not clear but it has successfully introduced to treat radiation-induced necrosis in several solid tumors [7,8].

To date, there is limited clinical evidence focusing on safety of anti-angiogenic drugs in pediatric patients. This is mainly due to the low prevalence and the heterogeneity of pediatric cancers. The purpose of this systematic review was to estimate the proportion of adverse events of anti-angiogenic drugs used to treat solid tumors in patients aged 0–18 years and to assess the potential knowledge gaps on safety of these drugs.

## 2. Materials and Methods

### 2.1. Study Design

This systematic review and meta-analysis was registered on PROSPERO website (CRD42022325182). This study was conducted according to the Preferred Reporting Items for Systematic Reviews and Meta-Analyses (PRISMA) [10].

### 2.2. Literature Search

We searched PubMed, ISI Web of knowledge and clinicaltrials.gov databases for retrieving the studies of interest. Articles that were published before 31 March 2022 were considered suitable for inclusion. The search strategy was composed by three sets of keywords related to the concepts “anti-VEGF/anti-angiogenic drugs”, “pediatric patients”, and “cancer” (the full strings are available in Appendix A). Snowballing search was also conducted to retrieve additional papers of interest by examining the references cited in the included articles and in the excluded reviews that were retrieved from search strategy.

### 2.3. Eligibility Criteria

Studies including pediatric patients (≤18 years old) with solid cancers were selected for inclusion in the systematic review. If the study included both adults and pediatric patients, the article was considered suitable for inclusion only if reported a stratification of adverse events per pediatric patients or if the median age of included subjects was less than 18 years old. Studies reporting the safety of anti-angiogenic drugs in Table 1 were considered suitable for inclusion. Given the low prevalence of solid tumors in pediatric patients and the difficulty in conducting large comparative studies in this population, both comparative and non-comparative clinical trials and observational studies were considered suitable for inclusion. Case report and case series were not included. Given the possible misclassification between case series and cohort design, we adopted the definition proposed by Mathes et al., where cohort studies were defined as studies where (1) there is a comparison group, (2) a relative risk can be calculated among different exposures, or (3) patients are sampled on the basis of exposure and not on the basis of disease or disease-related outcomes [11]. If one or more of these conditions were true, the study was considered a cohort study. To be included in the meta-analysis, studies had to report at least the severity (grade) or seriousness (serious or non-serious) of adverse events (AEs) occurred. Moreover, to allow for a more formal analysis and to calculate the proportion of AEs [12], only those studies reporting the number of patients experiencing AEs were considered. Finally, as for those studies referring to the same clinicaltrials.gov (NCT) identifier, if the included study reported only a sub-cohort analysis such study was excluded from the meta-analysis to avoid that the same patient was counted twice.

### 2.4. Study Selection

As for published articles retrieved from PubMed and ISI web of Science, two authors (A.S. and V.C.) screened all titles and abstracts of the references retrieved. Potentially relevant studies were further assessed through examination of full texts. To search for unpublished clinical trials, all records from clinicaltrials.gov with published results were also screened and assessed for inclusion. The reviewers worked independently, in parallel, and blinded to each other. Disagreement between the two reviewers was solved through discussion with a third author (S.D.). As for published studies, eligible studies had to be written in English and studies with no full-text available were excluded.

### 2.5. Data Extraction

The following information was extracted from both published studies and clinicaltrials.gov records:Study characteristics: study type (e.g., phase I, cohort study); as for observational studies, the nature of data collection (i.e., retrospective or prospective) and study design were also extracted (e.g., cohort and case–control studies). The selected studies were associated with an NCT identifier when available.Disease and patients’ characteristics: solid tumor type (e.g., glioma) with its respective stage, number of patients included in the safety analysis, number of females, median age (range), and median follow-up time (range) were extracted.Exposure: dose, treatment schedule, formulation, and combined regimens to anti-angiogenic drug.Adverse events (AEs): type (e.g., nausea, hypertension), severity (i.e., grades 1–2, ≥3), seriousness (serious, non-serious), and number of patients experiencing AEs retrieved from selected studies; AEs reported in the retrieved studies were also assigned to a system organ class (SOC) according to the common terminology criteria for adverse events (CTCAE) version 5.0. Dose limiting toxicities were excluded from extraction. If a study reported separately the number of patients experiencing grades 3 and 4 AEs (or grades 1 and 2), the higher number between those graded 3–4 (as well as 1–2) was extracted to avoid a patient having experienced more than one event. An adverse event (AE) was defined as an unfavorable outcome that occurs during or after the use of a drug but is not necessarily caused by it [13]. As for severity, we considered the CTCAE version 5.0 to identify AE grade [14]. On the other hand, as for seriousness, we followed the ICH E2A guidelines from European Medicine Agency [15].

Information was collected in a specific data sheet and was validated by a second author (V.C.). Disagreement between the two reviewers was solved through discussion with a third author (S.D.).

### 2.6. Quality Assessment

Quality of the studies eligible for meta-analysis was assessed on the basis of the its study design [16,17]: quality of comparative randomized clinical trials was assessed through the Cochrane risk of bias tool [18], and quality of non-randomized or non-comparative clinical trials was assessed through the methodological index for non-randomized studies (MINORS) tool [19]. As in previous systematic reviews [20,21], a quality assessment of non-comparative observational studies was not conducted because these studies were assumed to be associated with high risk of bias. As for the Cochrane risk of bias tool, seven items were considered, namely random sequence generation, allocation concealment, blinding of participants and personnel, blinding of outcome assessment, incomplete outcome data, selective reporting, and other biases, and items were judged as “high risk”, “low risk”, and “unclear risk”. The overall risk of bias will correspond to the worst risk of bias in any of the domains. Moreover, if a study has “some concerns” in multiple items, it will be judged also as high risk of bias.

### 2.7. Statistical Analysis

Characteristics of the studies were described by year of publication, study design, tumor type, number of patients included in the safety population, number of females, median follow-up, and type of therapy (as well as dose and concomitant drugs).

As for the meta-analysis, the proportion of AEs was defined as the number of patients experiencing AEs divided by the total number of patients receiving a single drug/combination of drugs in the included studies. Clinical trials and observational studies data were also analyzed separately.

Study proportions of AEs were pooled with the “metaprop” command in R software (R Foundation) as follows [22]: we fitted random intercept logistic regression model and used maximum-likelihood estimator for tau^2^, logit transformation of proportions, and Clopper–Pearson CI for individual studies. Pre-planned heterogeneity investigation was based on different combination of anti-angiogenic drugs (with chemotherapy/as monotherapy): as for chemotherapy, single combinations were also investigated, where a high heterogeneity was found. Heterogeneity was assessed by inspecting I^2^ (>75% high, 40–74% moderate, <40 low).

## 3. Results

### 3.1. Study Selection

In total 300, 792, and 112 records were identified from PubMed, ISI Web of Knowledge, and clinical trials.gov, respectively (Figure 1). After removing duplicated records, 1093 articles were available for the screening of title and abstract: 109 studies were selected for full-text assessment. Twenty-three were excluded due to missing full texts, eight did not reported any safety outcomes, seven did not included pediatric patients, three did not include patients with solid tumors, two were case series/reports, and two did not analyze anti-angiogenic drugs. Eight records were found through the snowballing procedure, and thirty-two records on *clincialtrials.gov* without available publication but with study results were found. After screening, six additional clinical studies with results on clinicaltrials.gov were included in the analysis. In total, 70 records (64 published articles and 6 clinical trials with published results retrieved from clinicaltrials.gov) reporting AEs of anti-angiogenic drugs in pediatric patients with solid tumors were included in the systematic review [23,24,25,26,27,28,29,30,31,32,33,34,35,36,37,38,39,40,41,42,43,44,45,46,47,48,49,50,51,52,53,54,55,56,57,58,59,60,61,62,63,64,65,66,67,68,69,70,71,72,73,74,75,76,77,78,79,80,81,82,83,84,85,86,87,88,89,90,91,92,93].

### 3.2. Study Characteristics

Characteristics of the included records were reported in Table 2. Twenty-four were phase I studies, twenty-two were phase II studies, fifteen were retrospective observational studies, five studies were phase I/II, four studies were clinical trials not otherwise specified, and one was a phase II/III trial for a total of 1837 subjects: the median of patients enrolled per study was 25 (range 2–92). Some publications included in the systematic review referred to the same trial but analyze different indications, outcomes or different drug combinations (i.e., NCT00381797 [33,75,79,80], NCT00665990 [34,85,86], and NCT02432274 [35,36]).

Most of the studies evaluated AEs of anti-angiogenic drugs in patients with central nervous system tumors (sixteen various glioma [24,25,29,32,43,44,45,46,65,75,76,80,81,82,83,84], eight various brain tumors [28,33,54,59,61,64,72,88], three ependymoma [30,70,79], two medulloblastoma [23,51], one astrocytoma [47], and two neuroblastoma [56,78]). Twenty-three studies referred to a cohort of patients with various solid tumors [27,31,34,37,38,39,40,41,50,53,55,58,60,63,66,68,71,73,74,85,86,87,89], and ten referred to patients with sarcoma [26,35,36,52,57,62,69,77,78,90]. Additionally, two studies on gastrointestinal tumor [42,67], two on hepatic carcinoma [48,92], one study on thyroid cancer [49], and one on bone tumors [91] were found.

Twenty studies reported the number of AEs of anti-angiogenic drugs used as monotherapy (three for bevacizumab [40,43,72], four for sorafenib [47,71,90,91], three for sunitinib [31,67,70], two for pazopanib [41,73], two for vandetanib [42,49], one for aflibercept [39], one for axitinib [37], one for cabozatinib [27], one for trebananib [50], one for lenvatinib [35], one for regorafenib [38], and one for ramucirumab [74]), thirty-six reported anti-angiogenic drugs used in combination with standard chemotherapy (twenty-nine for bevacizumab [23,26,28,29,33,44,45,46,51,52,54,55,56,57,58,64,66,68,75,76,77,78,79,80,81,82,83,84], four for sorafenib [48,53,60,92], two for pazopanib [62,69], and one for lenvatinib [36]), and ten reported a combination with different regimens (eight for bevacizumab [30,32,59,61,63,65,88,89] and two for vandetanib [24,25]). Finally, four records reported the combination between bevacizumab and sorafenib with cyclophosphamide [34,85,86,87]. See Figure 2 for the number studies retrieved per anti-angiogenic drug.

Only few studies reported information on median follow-up of patients (12 out of 70), and the maximum follow-up reported was 96 months in a phase I/II study of patients receiving vandetanib as monotherapy [49]. The minimum follow-up reported was 7 months [69].

### 3.3. Safety

Forty-five studies were included in the meta-analysis for the evaluation of severity, while twenty for the evaluation of seriousness of AEs. Characteristics of the safety population of the selected studies were reported in Appendix A.

#### 3.3.1. Anti-Angiogenic Drugs as Monotherapy

In Figure 3, the overall proportion of serious (Panel A) and non-serious AEs (Panel B) for anti-angiogenic drugs used as monotherapies was reported. Overall, the proportion of serious and non-serious AEs of anti-angiogenic drugs used as monotherapy was 0.46 [95% CI: 0.24–0.69] and 0.89 [95% CI: 0.73–0.96], respectively. The two drugs with the higher proportion of serious AEs were sunitinib (0.79: one study) and lenvatinib (0.64; one study).

As for bevacizumab, only one study with seven patients in the safety analysis was included [72]. No serious AEs were found in this study. As for vandetanib, the two studies included in this review reported a proportion of serious AEs of 0.27 [95% CI: 0.11–0.51], with low heterogeneity (I^2^: 0%; *p* > 0.05) [42,49]. As for sorafenib, the proportion was 0.48 but with high heterogeneity between studies (I^2^: 89%; *p* < 0.01). Indeed, the dosages of sorafenib were different between the three studies included in this review (Widemann et al.: 150–325 mg/m^2^/dose; Kim et al.: 200 mg/m^2^/dose; Karajannis et al.: 200–400 mg/m^2^/dose) [47,71,90]. Moreover, as for the safety populations, while the rate between male: female was 1:1 for the study of Widemann et al. and Karajannis et al., the study ok Kim et al. is composed almost of female patients (nine out of ten) [47,71,90].

As for drugs used as monotherapy reporting both serious and grade ≥ 3 AEs and which were evaluated in at least 20 patients, the proportion of such AEs was reported in Figure 4 (see Appendix A for the other drugs).

Overall, the proportion of grade ≥ 3 and serious AEs, such as gastrointestinal AEs, hematological AEs, thromboembolic event, intracranial hemorrhage, hypertension, hypothyroidism, AST and ALT increase, rush, and proteinuria remained under the threshold of 0.15 for each drug. Sunitinib reported a proportion of neutropenia grade ≥ 3 of 0.26 [95% CI: 0.17–0.39] despite the proportion of serious neutropenia being 0.01 [95% CI: 0.01–0.21]. Additionally, trebananib reported a proportion of neutropenia grade ≥ 3 of 0.21 [95% CI: 0.08–0.44], but no information on seriousness was available from retrieved studies (Appendix A). Sorafenib reported a proportion of serious anemia and lymphopenia of 0.13 [95% CI: 0.06–0.26] and 0.13 [95% CI: 0.05–0.31], respectively. Moreover, for serious intracranial hemorrhage, a proportion of 0.10 [95% CI: 0.03–0.28] was found for patients treated with sunitinib.

As for drugs used as monotherapy reporting both non-serious and grade < 3 AEs and which were evaluated in at least 20 patients, the proportion of such AEs was reported in Figure 5 (see Appendix A for the other drugs). For each drug reported in Figure 5, we found a substantial correspondence between proportion of non-serious and grade < 3 AE. A higher proportion of hematological AEs was found in the patients receiving pazopanib (proportion for leukopenia: 0.22 [95% CI: 0.14–0.35], thrombocytopenia 0.28 [95% CI: 0.18–0.41]) and sorafenib (proportion for leukopenia: 0.23 [95% CI: 0.07–0.54], thrombocytopenia 0.24 [95% CI: 0.13–0.40]) with respect to sunitinib (proportion for leukopenia: 0.03 [95% CI: 0.01–0.21], thrombocytopenia 0.02 [95% CI: 0.01–0.22]) and lenvatinib (proportion for leukopenia: 0.01 [95% CI: 0.01–0.12], thrombocytopenia 0.09 [95% CI: 0.04–0.20]). Notably, sunitinib showed the proportion of non-serious neutropenia of 0.24 [95% CI: 0.12–0.43]. On the other hand, lenvatinib reported the higher proportion of non-serious hypertension (0.36 [95% CI: 0.25–0.50]) and of hypothyroidism (0.47 [95% CI: 0.34–0.60]).

#### 3.3.2. Anti-Angiogenic Drugs in Combination with Standard Chemotherapy

Twenty-five out of twenty-six studies included in the seriousness analysis reported both the overall proportion of serious and non-serious AEs. In Appendix A, the overall proportion of serious (Panel A) and non-serious AEs (Panel B) for anti-angiogenic drugs in combination with chemotherapy was reported.

The proportion of serious and non-serious AEs of anti-angiogenic drugs in combination with chemotherapy was 0.51 [95% CI: 0.32–0.69] and 0.90 [95% CI: 0.80–0.96], respectively. In combination with chemotherapy the proportion of serious AEs was found to be 0.74 [95% CI: 0.58–0.86] with lenvatinib (only one study), and 0.48 [95% CI: 0.29–0.68] with bevacizumab (thirteen studies). For non-serious AEs, bevacizumab in combination with chemotherapy reported a proportion of 0.89 [95% CI: 0.77–0.95], while in the only study included for lenvatinib, the authors reported that all the patients have at least one non-serious AEs. However, a high heterogeneity was found between studies reporting serious and non-serious AEs for bevacizumab with chemotherapy: I^2^ 88% and 82%, respectively (Figure 6).

The combination with a higher proportion of serious AEs was bevacizumab + vincristine + ifosfamide (0.93; 95% CI: 0.84–0.98), while that with less proportion was bevacizumab + doxycycline + methotrexate: (0.07; 95% CI: 0.01–0.19). For non-serious AEs, a high heterogeneity was found for combination of bevacizumab + irinotecan + temozolomide (three studies: I^2^ 86%) and for bevacizumab + temozolomide (two studies: I^2^ 91%). As for the first combination, in the study of Modak et al. [56] (which reported the higher proportion of serious AEs), bevacizumab was administered at 15 mg/kg, while in the other two studies (Levy et al.; Metts et al. [51,54]), bevacizumab was administered at 10 mg/kg. Moreover, the in the study of Metts et al. [54], the irinotecan was administered only at day 1 and 15 of a 28-day cycle, while in the other two studies [51,56], it was administered for the first five days of each cycle. Finally, as for the combination bevacizumab + temozolomide, temozolomide was given as 150–200 mg/m^2^/dose. These patients also received daily radiotherapy treatment, but its dosage was not reported in one study [76].

#### 3.3.3. Anti-Angiogenic Drugs in Combination with Different Regimens

Eight studies reported AEs of anti-angiogenic drugs in combination with regimens other than standard chemotherapy (six for bevacizumab [30,32,59,61,63,65], and two for vandetanib [24,25]).

The study of Federico et al. reported AEs for the combination bevacizumab + sorafenib + cyclophosphamide [34]. The study did not reported information about the seriousness but provided results about severity of AEs. The authors reported a proportion of 0.46 [95% CI: 0.27–0.65], 0.71 [95% CI: 0.51–0.85], 0.29 [95% CI: 0.15–0.50], 0.13 [95% CI: 0.04–0.31], and 0 [95% CI: 0–0.14] for grade ≥ 3 AEs leukopenia, lymphopenia, neutropenia, thrombocytopenia, and anemia, respectively. Moreover, the authors reported proportions of grade ≥ 3 for proteinuria of 0.08 [95% CI: 0.02–0.26] and for hypertension of 0.17 [95% CI: 0.07–0.36].

One study reported the combination of bevacizumab with temsirolimus [59] and one with everolimus [63]. As for the combination with everolimus, proportions of grade ≥ 3 of 0.27 [95% CI: 0.10–0.52] for lymphopenia and of 0.13 [95% CI: 0.03–0.37] for neutropenia and thrombocytopenia were reported. No vomiting or diarrhea grade ≥ 3 AEs were observed. In the study of bevacizumab + temsirolimus (six patients), no hematological and gastrointestinal AEs grade ≥ 3 were observed, except for one event of thrombocytopenia. The safety of bevacizumab was studies also in combination with lapatinib (anti-HER2) [30]. In this study, only the seriousness of AEs was reported. Overall, a proportion of 0.45 [95% CI: 0.28–0.65] was found for serious AEs. The authors reported proportions of 0.04 [95% CI: 0.01–0.20] for serious febrile neutropenia and increase AST and of 0.12 [95% CI: 0.04–0.31] for increase ALT. All patients had at least one non-serious AEs (proportion for diarrhea: 0.75 [95% CI: 0.55–0.88]; vomiting 0.33 [95% CI: 0.18–0.53]; hypertension: 0.13 [95% CI: 0.04–0.31]).

One study by Su et al. reported information for safety of bevacizumab + radiotherapy + valproic acid in pediatric patients with glioma [65]. Serious AEs occurred in the 10% of the safety population. No hematological and gastrointestinal serious AEs were found. Non-serious AEs occurred in the 87% of the safety population and the proportion of non-serious hypertension was 0.32 [95% CI: 0.19–0.47].

Finally, two studies evaluated severity of AEs for vandetanib: one study in combination with radiotherapy + dasatinib [25], and the other study with the only radiotherapy [24]. The study of Broniscer et al., 2013 (vandetanib + dasatinib + radiotherapy) reported proportions of grade ≥ 3 for neutropenia of 0.16 [95% CI: 0.06–0.34] and for anemia of 0.08 [95% CI: 0.02–0.24]. The proportion of grade ≥ 3 for diarrhea was 0.08 [95% CI: 0.02–0.24] [25]. The study of Broniscer et al., 2010 (vandetanib + radiotherapy) found proportions of grade ≥ 3 for lymphopenia of 0.69 [95% CI: 0.52–0.81] and for neutropenia of 0.09 [95% CI: 0.03–0.22] [24].

### 3.4. Quality of Included Studies

We found seven randomized studies out of 52 clinical trials (13%) included in the meta-analysis that were evaluated with the Cochrane Risk of Bias tool. All randomized studies were open label and were considered as a high risk of bias for both blinding of participants/personnel and outcome. The other included clinical trials were not comparative and were evaluated with the MINORS tool. Twenty-eight studies reported a moderate quality, seven reported a good quality, and seven reported a poor quality. Appendix A (risk of bias tool) and Appendix A (MINORS tool) reported the quality of the included studies. 

## 4. Discussion

This is the most comprehensive systematic review and meta-analysis aimed to assess the safety of anti-angiogenic drugs in pediatric population with solid cancer. Anti-angiogenic drugs are widely used in childhood cancers, although none of these are approved in pediatric oncology. However, this study did not take in consideration those drugs with anti-angiogenic proprieties that are proposed to be repurposed for solid malignancies (i.e., propranolol and sirolimus) in adults [94,95]. Overall, our review found 70 articles and our meta-analysis is based on 56 studies, which included about 1500 patients, since 2008 to 2022.

Most anti-angiogenic treatments targeting VEGF in pediatric population have been investigated up to a phase II study. We found only one phase II/III trial in which bevacizumab safety in combination with temozolomide and radiotherapy was assessed (NCT01236560) [76]. Bevacizumab was the most investigated drug in the pediatric oncology population, followed by sorafenib.

In our analysis, lenvatinib and sunitinib showed the higher proportion of serious AEs in monotherapy. However, both lenvatinib and sunitinib evidence was drawn by two single studies (55 patients for lenvatinib; 29 patients for sunitinib).

The results from our review showed that gastrointestinal as well as hematological events were the most common AEs in patients receiving anti-angiogenic drugs in monotherapy, despite the proportion of serious/severe remained under 0.15 for each drug (except for sunitinib, which reported a severe neutropenia proportion of 0.27 [CI 95%: 0.17–0.40]). This higher proportion of severe neutropenia reported for sunitinib might be explained by the direct toxicity of multi-TKIs on hematopoietic progenitor cells [96].

Overall, the frequency and severity of myelosuppression vary among anti-angiogenic drugs, based on their different anti-kinase selectivity (lenvatinib reported the lowest rate of proportion of hematological AEs—see Figure 4 and Figure 5). Their activity against fms-related tyrosine kinase 3 (FLT3 or CD135) and c-kit, which are essential for survival and differentiation of hemopoietic progenitor cells, is critical to determine their hematologic toxicity profiles [97]. A possible mechanism might also involve ROS generation, related to both their efficacy and toxicity [98].

In agreement with what observed in adult cancer population, the most common cardiovascular event in pediatric cancer patients exposed to antiangiogenic drugs was hypertension [41,69,70]. The proportion of serious thromboembolic event and intracranial hemorrhage was found less than 5% for each drug except for sunitinib (0.10 CI 95%: 0.17–0.40) However, one of two studies assessing sunitinib for serious AEs was conducted on patients with ependymoma (brain tumor), which need to be considered as a confounding factor for the occurrence of this event. Several mechanisms were suggested for the association between VEGF signaling inhibition and the development of cardiovascular events. VEGF induces the production of two vasodilators, nitric oxide, and prostacyclin, as well as inhibits the production of the vasoconstrictor endothelin-1 [99]. In addition, VEGF promotes proliferation and inhibits apoptosis of endothelial cells, thus contributing to maintenance of vascular homeostasis and tumor angiogenesis [100].

Interestingly, the results from our study showed that pediatric patients treated with lenvatinib reported a high proportion of non-serious hypothyroidism (0.47 CI 95%: 0.34–0.60). This AE associated with lenvatinib is also reported in adult patients affected by hepatocarcinoma, with a proportion varying from 0.16–0.21 [101,102]. However, the mechanism by which lenvatinib induce thyroiditis is not clear. Two studies reported that TKIs induce hypothyroidism through tissue ischemia (inhibition of thyroid blood flow) or apoptosis of the thyroid follicular cells [103,104]. Unfortunately, this systematic review retrieved only seven observational studies (six for bevacizumab, one for pazopanib) that meet the inclusion criteria for meta-analysis [23,28,29,46,55,61,62], and given the different regimens analyzed, indications and doses these studies were not pooled together and no conclusive evidence on the long-term safety of anti-angiogenic drugs in pediatric patients could possibly be drawn. This aspect is important in light of preclinical and clinical evidence on adults patients: in particular, pazopanib is reported to increase risk for bone shortening and fragility, and tooth remodeling (the effects are reported in young rats at ≥10 mg/kg/day) as well as evidence coming from some TKIs, which are associated with cardiovascular events [105,106]. Thus, while the use of pazopanib is not recommended in patients <2 years of age due to safety concerns related to growth and organ maturation [107], that of anti-angiogenic TKIs is currently not regulated due to lack of clear data on safety. Data from observational studies, as well as pharmacovigilance studies, could help to define mechanism of action-depending toxicities of newer multi-TKIs and to evaluate the long-term safety of anti-angiogenic drugs in pediatric patients.

Finally, as for serious and non-serious AEs of anti-angiogenic drugs used in monotherapy, this study highlighted a high heterogeneity among the studies reporting on sorafenib. In the three studies on sorafenib, the drug was used with different posology. Unfortunately, exclusion and inclusion criteria presented by single protocols for each study assessing sorafenib in monotherapy were not retrieved, and we are not able to provide any conclusive assumptions on the heterogeneity observed.

As expected, the combination of anti-angiogenic drugs with chemotherapy leads to an increase in AEs compared with monotherapy. Regarding serious Aes, the combination of bevacizumab with vincristine, ifosfamide, and doxorubicin (only one study) showed the worst safety profile, reaching a proportion of 0.93 [CI 95%: 0.84–0.98]. On the other hand, bevacizumab in combination with doxycycline and methotrexate exhibited the better safety profile (serious AEs 0.07; CI 95%: 0.01–0.19).

The various proportion of serious AEs seemed to be due by the different standard chemotherapy regimen and its different posology. To confirm this observation, among different groups of standard chemotherapy regimens, we did not observe a high heterogeneity regarding serious AEs. As for non-serious AEs, a high heterogeneity was found for the group of bevacizumab + temozolomide and in the group of bevacizumab + irinotecan + temozolomide. A possible explanation of this heterogeneity could be related to different reporting/monitoring of non-serious AEs between the studies.

Although in the adult population, there are no indications on the efficacy of the bevacizumab and sorafenib combination and a clear evidence of excessive toxicity [108,109], in pediatric patients, one study investigated this combination (phase II study with 44 patients with solid tumors) [34]. We found a high proportion of grade ≥ 3 hematological AEs for this combination (proportion of leukopenia 0.46, lymphopenia 0.71, and neutropenia 0.29). In the literature, two other studies, not included in the systematic review due to eligibility criteria, reported information on the safety of bevacizumab in combination with sorafenib in pediatric patients [85,87]. Two safety notes could be derived from such studies: patients with lung lesions and dermatological lesions receiving bevacizumab and sorafenib should be carefully monitored for signs and symptoms of pneumothorax and hand-foot syndrome, respectively. Both toxicities have been described in trials evaluating anti-angiogenic agents [86,110].To our knowledge, this is the first systematic review and meta-analysis assessing the safety of anti-angiogenic drugs in pediatric patients with solid tumors. We conducted a systematic review and meta-analysis according to PRISMA guidelines, including more than 50 clinical studies. To allow a more formal analysis of proportion of AEs we included only studies reporting number of patients with AEs and we minimized retrieved available literature by also including studies reporting results on clinicaltrials.gov.

However, this study also has some limitations. First, only seven studies were randomized clinical trials and most of the included studies were non-comparative clinical trials (42 studies). Given the low prevalence of solid tumors in pediatric patients and the difficulties in conducting large comparative trials in such population, the non-comparative nature of the most of the studies was expected. Second, as for non-comparative trials, most of them were found to have a moderate quality (28/42) and seven were found to have low quality, while clinical trials were open label and so each of them was considered as a high risk of blinding of participants, personnel, outcome assessment. Third, in this work, we decided not to report pharmacokinetic parameters to reveal the safety profile of anti-angiogenic drugs in relation to plasma concentration because it was outside the scope of this review. This systematic review could represent a starting point for the evaluation of pharmacokinetics parameters on safety of anti-angiogenic drug use in pediatric patients. Additionally, gender may affect the metabolic activity of enzymes involved in pharmacokinetics of pro-angiogenic drugs. Nevertheless, despite gender seemingly playing a role in the survival of patients exposed to anticancer drugs [111,112], the role of sex and gender on safety for anticancer drug use, in particular, for anti-angiogenic drugs, is still controversial (especially in the pediatric population). Notably, none of the study included in this review discussed the tolerability results of anti-angiogenic drugs with respect to sex and gender. Finally, it should be considered that selective reporting is also very frequent when dealing with reviews of adverse events. The observation period of the included studies in the meta-analysis was very heterogeneous (most of them did not report it), and the probability of observing events for longer studies is higher than that for short studies.

According to the results of this systematic review and meta-analysis, we can also provide some recommendations. First, we did not obtain any information on the long-term toxicities, and observational studies with long-term follow-up using routinely collected electronical healthcare data are required. Second, this study did not aim to collect evidence from studies using spontaneous reporting system; however, the use of these platforms could be fundamental to filling the gap in the safety of anti-angiogenic drugs in special populations.

## 5. Conclusions

In conclusion, anti-angiogenic drugs are frequently used in cancer childhood, from bevacizumab to the more recent molecularly targeted agents. Their toxicity profiles have mostly been studied in phase I and II trials and are limited to acute toxicities, while observational studies are limited on few drugs, such as bevacizumab and pazopanib.

Overall, we observed a correlation between seriousness and severity of AEs. Among monotherapy TKIs, the drugs with a higher proportion of serious and non-serious AEs were sunitinib and lenvatinib, while sorafenib reported a high heterogeneity among studies included. As expected, the proportion of AEs varied in relation to the single combination of anti-angiogenic drugs with standard chemotherapy or other targeted therapies/radiotherapy.

Currently, growth and developmental toxicity, such as those related to TKIs, still remain inadequately addressed. Data from observational studies could help to define mechanism of action-depending toxicities of newer multi-TKIs and to evaluate the long term safety of anti-angiogenic drugs in pediatric patients.

## Figures and Tables

**Figure 1 cancers-14-05315-f001:**
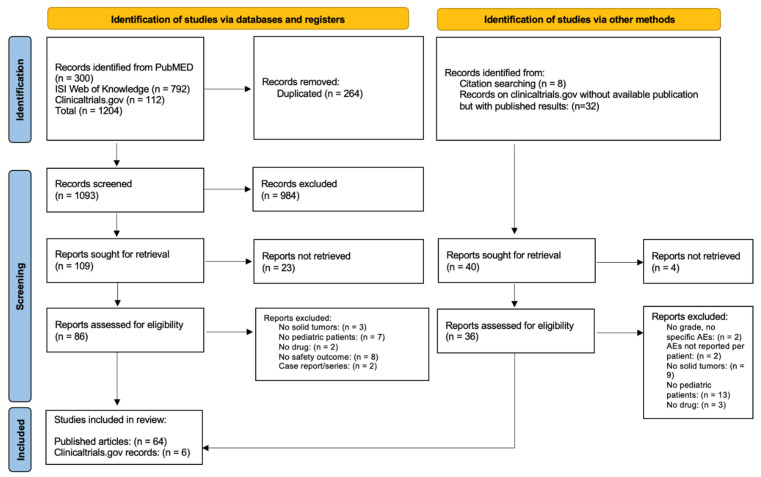
Prisma flow diagram.

**Figure 2 cancers-14-05315-f002:**
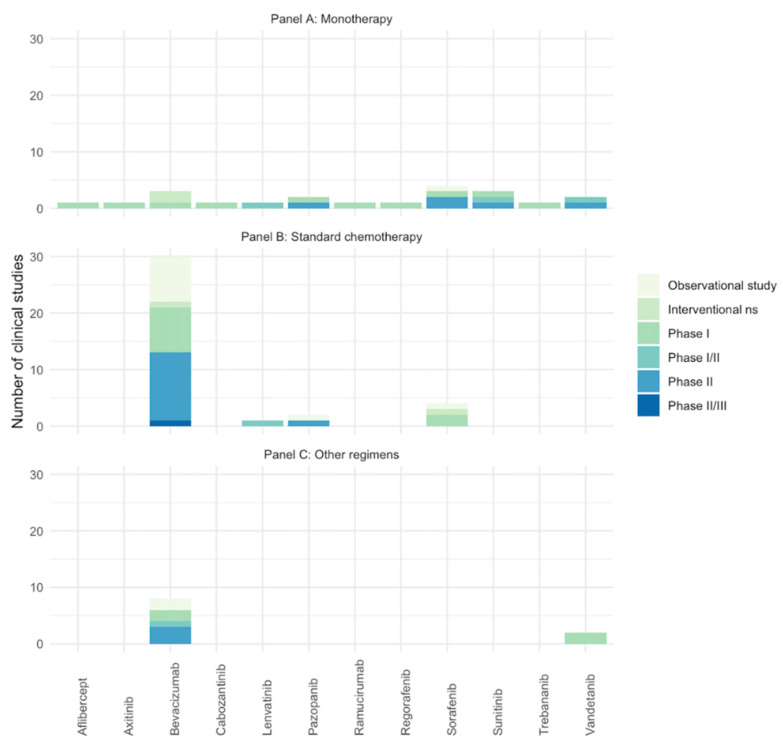
Number of included studies (clinical trials and observational studies) assessing the AEs of anti-angiogenic drugs in pediatric patients with solid tumors. Studies with same NCT identifier were considered as one. (**A**) Anti-angiogenic drugs in monotherapy; (**B**) anti-angiogenic drugs in combination with standard chemotherapy; (**C**) anti-angiogenic drugs in combination with target therapy/other. Ns: phase non-specified.

**Figure 3 cancers-14-05315-f003:**
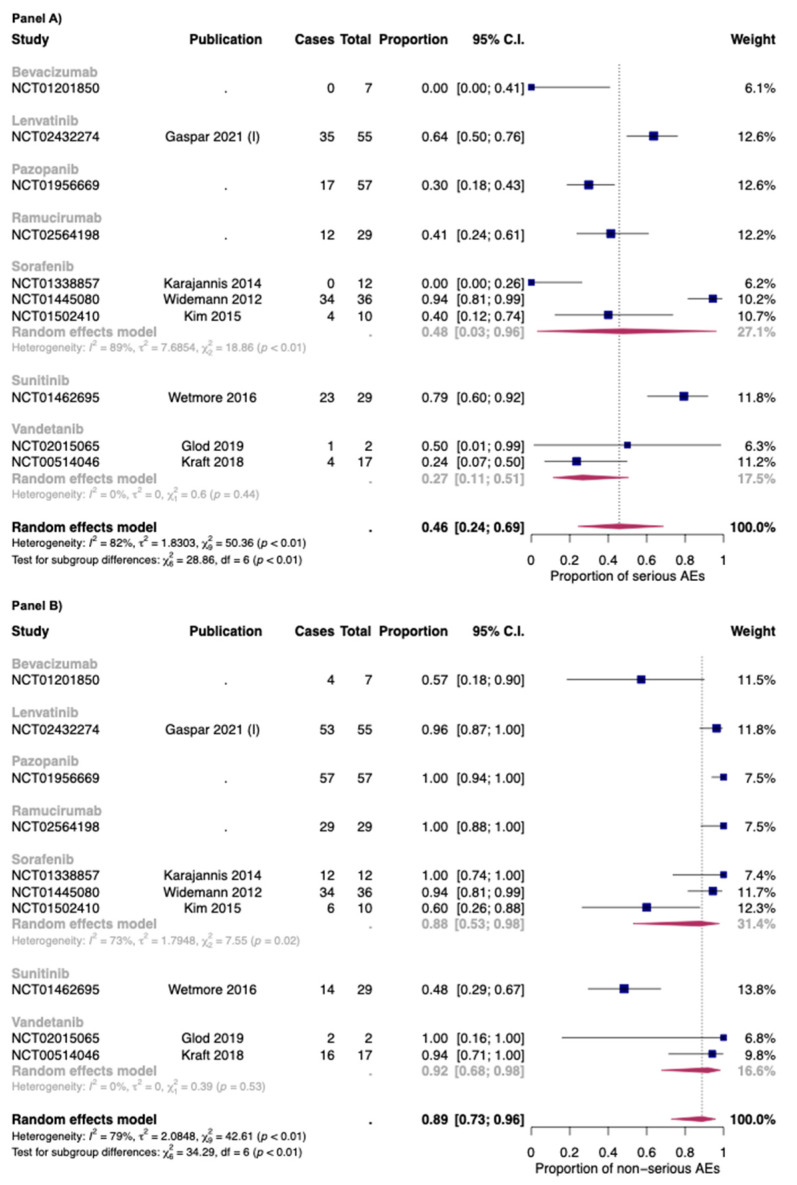
Overall proportion of serious (Panel **A**) and non-serious (Panel **B**) AEs for anti-angiogenic drugs used as monotherapy. Cases: Patients with at least an event. Total: Patients included in the study [35,42,47,49,70,71,90].

**Figure 4 cancers-14-05315-f004:**
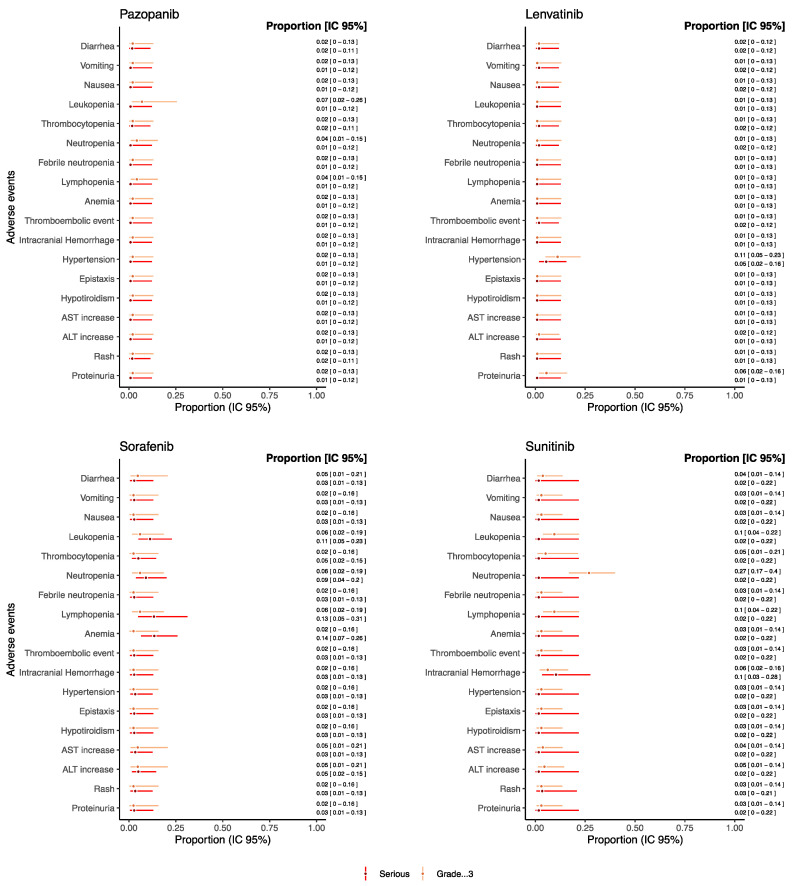
Proportions of serious/grade ≥ 3 AEs for pediatric patients receiving anti-angiogenic drugs as monotherapies.

**Figure 5 cancers-14-05315-f005:**
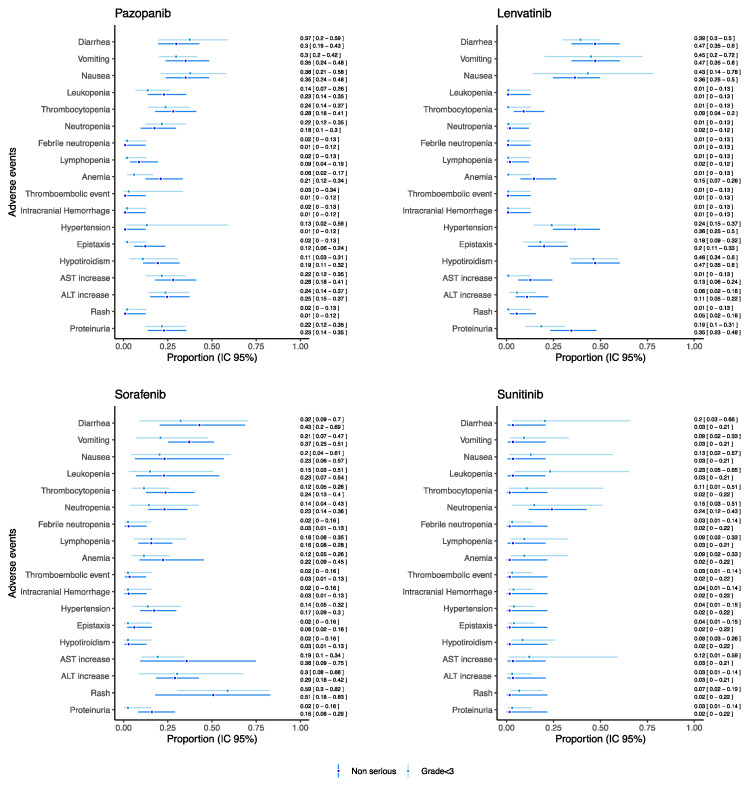
Proportions of non-serious/grade < 3 AEs for pediatric patients receiving anti-angiogenic drugs as monotherapies.

**Figure 6 cancers-14-05315-f006:**
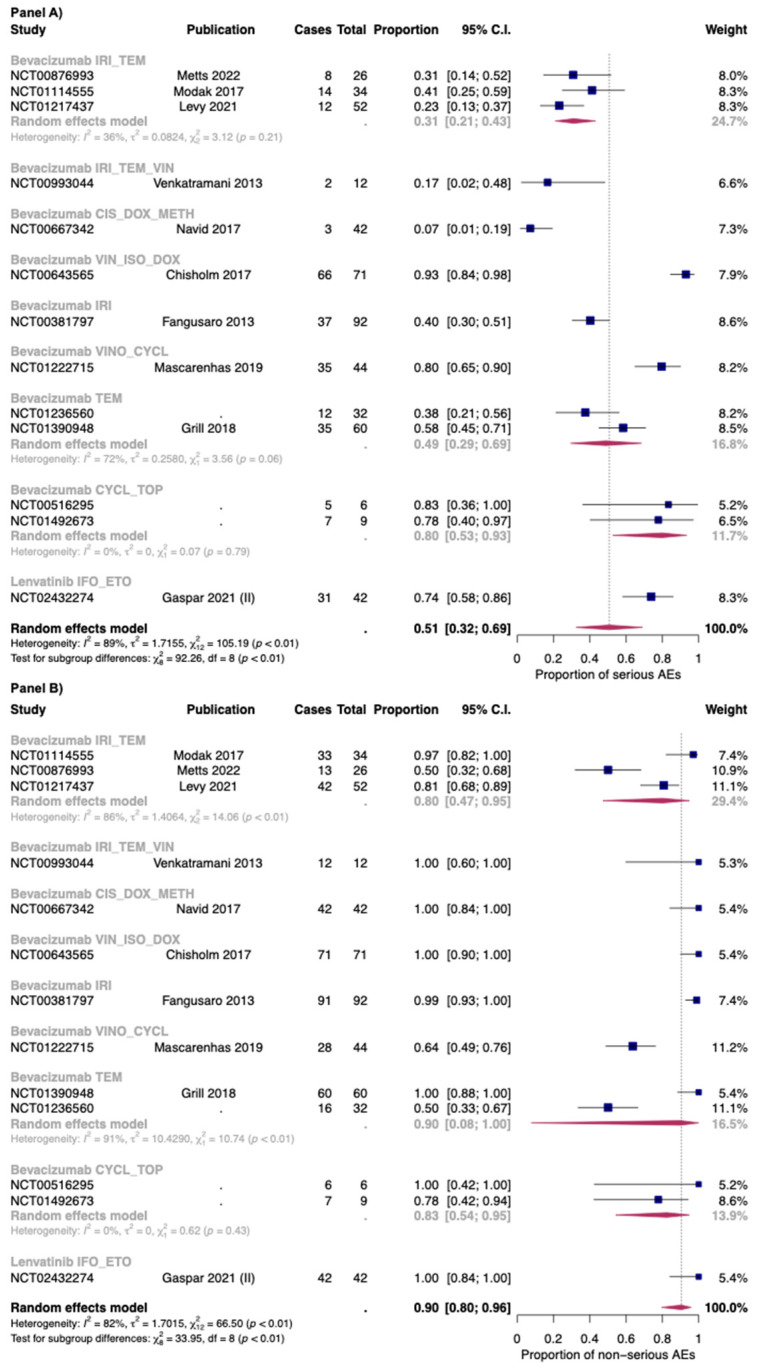
Proportion of serious (Panel **A**) and non-serious AEs (Panel **B**) for bevacizumab and lenvatinib plus chemotherapy by single regimens; IRI_TEM: Irinotecan + temozolomide; IRI_TEM_VIN: Irinotecan + temozolomide + vincristine; CIS_DOX_METH: Cisplatin + doxorubicin + methotrexate; VIN_ISO_DOX: Vincristine + ifosfamide +doxorubicin; IRI: irinotecan; VINO_CYCL: Vinorelbine + cyclophosphamide; TEM: Temozolomide; CYCL_TOP: Cyclophosphamide + topotecan; IFO_ETO: Ifosfamide + etoposide. Cases: Patients with at least an event. Total: Patients included in the study [26,33,36,44,51,52,54,56,57,66].

**Table 1 cancers-14-05315-t001:** Approved drugs with anti-angiogenic properties (known/potential) in patients with solid tumors.

Drug and Approval Year in Europe	Class	Target	Approved Clinical Use
Aflibercep (2012)	Soluble recombinant fusion protein	VEGF A-D, PlGF	Metastatic CRC (with FOLFIRI)
Axitinib (2012)	Multi-TKI	VEGFR-1/2/3	RCC, plus avelumab or pembrolizumab as the first treatment in advanced RCC
Bevacizumab (2005)	Humanized monoclonal antibody	VEGF-A	Metastatic CC, recurrent glioblastoma, HCC, advanced non squamous NSCLC, OEC, fallopian tube/primary peritoneal cancer, RCC
Cabozantinib (2014)	Multi-TKI	VEGFR-2, c-Met, ROS1, TYRO3, MER, Ret, Kit, TRKB Flt-3, AXL, Tie-2	Advanced RCC, HCC
Lenvatinib (2015)	Multi-TKI	VEGFR1/2/3, FGFRs, PDGFR-α, c-Kit receptor, RET	Metastatic RCC, RAI-DTC, HCC.
Nintedanib (2014)	Multi-TKI	VEGFR-1/2/3, PDGFR-α/β, FGFR-1/2	NSCLC (with Docetaxel therapy)
Pazopanib (2010)	Multi-TKI	VEGF-R-1/2/3, PDGF-R-α/β, c-Kit	Advanced RCC, advanced STS
Ponatinib (2013)	Multi-TKI	VEGF-R, SRC, ABL, FGF-R, PDGF-R.ABL-T135I mutation	CML, Ph^+^ ALL
Ramucirumab (2014)	Humanized monoclonal antibody	VEGFR-2	CRC, NSCLC, HCC, GC or GEJ adenocarcinoma
Regorafenib (2013)	Multi-TKI	VEGFR-1/2/3, PDGFR-α/β, FGFR-1/2, Tie2, RAF-1, BRAF, BRAF^V600E^ c-Kit receptor	mCRC, advanced GIST, HCC
Sorafenib (2006)	Multi-TKI	VEGFR-1/2/3, PDGFR-β, Raf serine/threonine kinases, c-Kit receptor	HCC, RCC, RAI-DTC
Sunitinib (2006)	Multi-TKI	VEGFR-1/2/3, PDGFR-α/β, c-Kit receptor, RET, FLT3, CSF-1R	Imatinib-resistant GIST, RCC, pNET
Tivozanib (2017)	TKI	VEGFR-1/2/3	Relapsed or refractory RCC
Trebananib (2013) *	Peptide-Fc fusion protein	Angiopoietins-1/2	Ovarian cancer
Vandetanib (2012)	Multi-TKI	VEGFR-2, EGFR, RET signaling	MTC

BC: breast cancer; BRAF: v-raf murine sarcoma viral oncogene homolog B1; CC: cervical cancer; CML: chronic myeloid leukemia; CRC: colorectal cancer; CSF-1R: colony stimulating factor 1 receptor; DFSP: dermatofibrosarcoma protuberans; DTC: refractory thyroid cancer; EGFR: epithelial growth factor receptor; EOC: epithelial ovarian cancer; FGFR: fibroblast growth factor receptor; FLT3: Fms-like tyrosine kinase 3; GC: gastric cancer; GEJ: esophagogastric junction; GIST: gastrointestinal stromal tumors; HCC: hepatocellular carcinoma; mCRC: metastatic colorectal cancer; MDS/MDP: myelodysplastic/myeloproliferative neoplasms; MTC: medullary thyroid cancer; NSCLC: non-small cell-lung cancer; RAF-1: v-raf-1 murine leukemia viral oncogene homolog 1; RCC: renal cell carcinoma; RET: Rearranged during transfection gene; ROS-1: ROS proto-oncogene 1; PDGF: platelet-derived growth factor; PDGF: platelet-derived growth factor receptor; Ph+ ALL: Philadelphia chromosome–positive acute lymphoblastic leukemia; PlGF: placentar growth factor; pNET: pancreatic neuroendocrine tumor; STS: soft tissue sarcoma; Tie-2: angiopoietin-2 receptor; TKI: tyrosine kinase inhibitors; VEGF: vascular endothelial growth factor, VEGFR: vascular endothelial growth factor receptor. * FDA approval date.

**Table 2 cancers-14-05315-t002:** Characteristics of included studies assessing safety of anti-angiogenic drugs in pediatric patients with solid tumors.

Reference	NCT Identifier	Study Type	Randomization	Pathology Characteristics	Anti-Angiogenic Drug	Dose	Concomitant Therapy
Glade Bender et al., 2008 [40]	-	Phase I	No	Solid tumors	Bevacizumab	5–15 mg/kg	No
Gorsi et al., 2018 [43]	-	Clinical trial ns	No	Glioma	Bevacizumab	10 mg/kg	No
-	NCT01201850	Clinical trial ns	No	CNS tumors	Bevacizumab	10 mg/kg	No
Okada et al., 2013 [58]	-	Phase I	No	Solid tumors	Bevacizumab	10 mg/kg	Irinotecan
Fangusaro et al., 2013 [33]	NCT00381797	Phase II	No	CNS tumors	Bevacizumab	10 mg/kg	Irinotecan
Gururangan et al., 2010 [75]	NCT00381797	Phase II	No	Glioma	Bevacizumab	10 mg/kg	Irinotecan
Gururangan et al., 2012 [79]	NCT00381797	Phase II	No	Ependymoma	Bevacizumab	10 mg/kg	Irinotecan
Gururangan et al., 2014 [80]	NCT00381797	Phase II	No	Glioma	Bevacizumab	10 mg/kg	Irinotecan
Couec et al., 2012 [28]	-	Observational study	-	Brain tumors	Bevacizumab	10 mg/kg	Irinotecan
Kalra et al., 2015 [46]	-	Observational study	-	Glioma	Bevacizumab	10 mg/kg	Irinotecan
De Marcellus et al., 2022 [29]	-	Observational study	-	Glioma	Bevacizumab	10 mg/kg	Irinotecan
Packer et al., 2009 [81]	-	Observational study	-	Glioma	Bevacizumab	10 mg/kg	Irinotecan
Modak et al., 2017 [56]	NCT01114555	Phase II	No	Neuroblastoma	Bevacizumab	15 mg/kg	Irinotecan + Temozolomide
Levy et al., 2020 [51]	NCT01217437	Phase II	Yes	Medulloblastoma	Bevacizumab	10 mg/kg	Irinotecan + Temozolomide
Hummel et al., 2016 [45]	NCT00890786	Phase I	No	Glioma	Bevacizumab	10 mg/kg	Irinotecan + Temozolomide
Metts et al., 2022 [54]	NCT00876993	Phase I	No	CNS tumors	Bevacizumab	10 mg/kg	Irinotecan + Temozolomide
Schiavetti et al., 2019 [64]	-	Clinical trial ns	No	Brain tumors	Bevacizumab	10 mg/kg	Irinotecan + Temozolomide
Aguilera et al., 2013 [23]	-	Observational study	-	Medulloblastoma	Bevacizumab	10 mg/kg	Irinotecan + Temozolomide
Crotty et al., 2020 [82]	-	Observational study	-	Glioma	Bevacizumab	10 mg/kg	Irinotecan + Temozolomide
Wagner et al., 2013 [68]	NCT00786669	Phase I	No	Solid tumors	Bevacizumab	15 mg/kg	Irinotecan + Temozolomide + Vincristine
Venkatramani et al., 2013 [66]	NCT00993044	Phase I	No	Solid tumors	Bevacizumab	15 mg/kg	Irinotecan + Temozolomide + Vincristine
El-Khouly et al., 2021 [32]	-	Phase I/II	No	Glioma	Bevacizumab	10 mg/kg	Irinotecan + Erlotinib
Grill et al., 2018 [44]	NCT01390948	Phase II	Yes	Glioma	Bevacizumab	10 mg/kg	Temozolomide + radiotherapy
-	NCT01236560	Phase II/III	Yes	Glioma	Bevacizumab	10 mg/kg	Temozolomide + radiotherapy
-	NCT01492673	Phase II	No	Ewing Sarcoma and Neuroblastoma	Bevacizumab	15 mg/kg	Cyclophosphamide + Topotecan
-	NCT00516295	Phase II	Yes	Ewing Sarcoma	Bevacizumab	ns	Cyclophosphamide + Topotecan + Vincristine
Mascarenhas et al., 2019 [52]	NCT01222715	Phase II	Yes	Rhabdomyosarcoma	Bevacizumab	15 mg/kg	Vinorelbine + Cyclophosphamide
Navid et al., 2017 [57]	NCT00667342	Phase II	No	Osteosarcoma	Bevacizumab	15 mg/kg	Methotrexate + doxorubicin + cisplatin
Chisholm et al., 2017 [26]	NCT00643565	Phase II	Yes	Sarcoma (soft tissue)	Bevacizumab	7.5 mg/kg	Vincristine, ifosfamide, actinomycin-D and doxorubicin
Zhukova et al., 2018 [83]	-	Observational study	-	Glioma	Bevacizumab	10 mg/kg	Chemotherapy
Parekh et al., 2011 [84]	-	Observational study	-	Glioma	Bevacizumab	10 mg/kg	Chemotherapy
Millan et al., 2016 [55]	-	Observational study	-	Solid tumors and vascular anomalies	Bevacizumab	Median dose: 9.25 mg/kg	+/− Chemotherapy
Piha-Paul et al., 2014 [59]	NCT00610493	Phase I	No	CNS tumors	Bevacizumab	5–10–15 mg/kg	Temsirolimus
Federico et al., 2020 [34]	NCT00665990	Phase I	No	Solid tumors	Bevacizumab	15 mg/kg	Sorafenib + Cyclophosphamide
Navid et al., 2012 [85]	NCT00665990	Phase I	No	Solid tumors	Bevacizumab	5 mg/kg	Sorafenib + Cyclophosphamide
Inaba et al., 2019 [86]	NCT00665990	Phase I	No	Solid tumors	Bevacizumab	15 mg/kg	Sorafenib + Cyclophosphamide
Interiano et al., 2015 [87]	-	Observational study	-	Solid tumors	Bevacizumab	15 mg/kg	Sorafenib + Cyclophosphamide
Su et al., 2020 [65]	NCT00879437	Phase II	No	Glioma	Bevacizumab	10 mg/kg	Valproic acid + radiotherapy
Santana et al., 2020 [63]	NCT00756340	Phase I	No	Solid tumors	Bevacizumab	8–10 mg/kg	Everolimus
De Wire et al., 2015 [30]	NCT00883688	Phase II	No	Ependymoma	Bevacizumab	10 mg/kg	Lapatinib
Peyrl et al., 2012 [88]	NCT01356290	Phase II	-	Brain tumors	Bevacizumab	10 mg/kg	Various
Reismuller et al., 2010 [61]	-	Observational study	-	CNS tumors	Bevacizumab	10 mg/kg	Various
Benesch et al., 2007 [89]	-	Observational study	-	Solid tumors	Bevacizumab	5–10 mg/kg	Various
Widemann et al., 2012 [71]	NCT01445080	Phase I	No	Solid tumors	Sorafenib	150–325 mg/m^2^/dose	No
Karajannis et al., 2014 [47]	NCT01338857	Phase II	No	Astrocytoma	Sorafenib	200 to 400 mg/m^2^/dose	No
Kim et al., 2015 [90]	NCT01502410	Phase II	No	Rhabdomyosarcoma and Wilms tumors	Sorafenib	200 mg/m^2^/dose	No
Raciborska et al., 2018 [91]	-	Observational study	-	Bone tumors	Sorafenib	100–400 mg/m^2^/dose	No
Meany et al., 2021 [53]	NCT01518413	Phase I	No	Solid tumors	Sorafenib	105–200 mg/m^2^	Irinotecan
Keino et al., 2020 [48]	-	Clinical trial ns	No	Hepatic cancer	Sorafenib	200–400 mg/m^2^/dose	Irinotecan
Reed et al., 2016 [60]	NCT01683149	Phase I	No	Solid tumors	Sorafenib	150 mg/200 mg	Topotecan
Schmid et al., 2012 [92]	-	Observational study	-	Hepatocellular carcinoma	Sorafenib	244–602 mg/m^2^/day	Cisplatin + doxorubicin
DuBois et al., 2011 [31]	NCT00387920	Phase I	No	Solid tumors	Sunitinib	15-20 mg/m^2^	No
Wetmore et al., 2016 [70]	NCT01462695	Phase II	No	Ependymoma	Sunitinib	15 mg/m^2^	No
Verschuur et al., 2019 [67]	NCT01396148	Phase I/II	No	Gastrointestinal tumor	Sunitinib	7.5–30 mg/m^2^	No
Glade Bender et al., 2013 [41]	NCT00929903	Phase I	No	Solid tumors	Pazopanib	275 to 600 mg/m^2^ (tablet); 50 mg/mL (suspension)	No
-	NCT01956669	Phase II	No	Solid tumors	Pazopanib	225–450 m^2^/m^2^/dose	No
Weiss et al., 2020 [69]	NCT02180867	Phase II	Yes	Sarcoma (soft tissue)	Pazopanib	7–5 g/m^2^	Doxorubicin, Ifosfamide
Russo et al., 2020 [62]	-	Observational study	-	Sarcoma	Pazopanib	450 mg/m^2^	Vincristine + irinotecan
Broniscer et al., 2010 [24]	NCT00472017	Phase I	No	Glioma	Vandetanib	50–145 mg/m^2^	Radiotherapy
Broniscer et al., 2013 [25]	NCT00996723	Phase I	No	Glioma	Vandetanib	65–85 mg/m^2^	Dasatinib + radiotherapy
Kraft et al., 2018 [49]	NCT00514046	Phase I/II	No	Thyroid Carcinoma	Vandetanib	100–300 mg/m^2^/dose	No
Fox et al., 2013 [93]	NCT00514046	Phase I/II	No	Thyroid Carcinoma	Vandetanib	100–300 mg/m^2^/dose	No
Glod et al., 2019 [42]	NCT02015065	Phase II	No	Gastrointestinal tumor	Vandetanib	100 mg/m^2^	No
Geoerger et al., 2021 [38]	NCT02085148	Phase I	No	Solid tumors	Regorafenib	60–93 mg/m^2^	No
-	NCT02564198	Phase I	No	Solid tumors	Ramucirumab	8–12 mg/kg	No
Leary et al., 2017 [50]	NCT01538095	Phase I	No	Solid tumors/tumors of the central nervous system	Trebananib	10/15/30 mg/kg	No
Chuk et al., 2018 [27]	NCT01709435	Phase I	No	Solid tumors	Cabozantinib	30–55 mg/m^2^/day	No
Geller et al., 2018 [37]	NCT02164838	Phase I	No	Solid tumors	Axitinib	2.4 and 3.2 mg/m^2^/dose	No
Glade Bender et al., 2012 [39]	NCT00622414	Phase I	No	Solid tumors	Aflibercept	2.0, 2.5 or 3.0 mg/kg	No
Gaspar 2021 et al. (I) [35]	NCT02432274	Phase I/II	No	Solid tumors (phase I), osteosarcoma (phase II)	Lenvatinib	11–17 mg/m^2^	No
Gaspar 2021 et al. (II) [36]	NCT02432274	Phase I/II	No	Osteosarcoma	Lenvatinib	11–14 mg/m^2^	Ifosfamide + etoposide

CNS: Central nervous system; Ns: Not specified; -: information missing or not found.

## Data Availability

The data can be shared upon request.

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
