# Peer review of "Safety of Anti-Angiogenic Drugs in Pediatric Patients with Solid Tumors: A Systematic Review and Meta-Analysis"

_cancers, 2022, doi:10.3390/cancers14215315_

Round 1

Reviewer 1 Report

Andrea Spini and colleagues performed a systematic review and meta-analysis on the safety of antiangiogenic drugs either as monotherapy or in combinations in pediatric patients with solid tumors.  Total of 70 studies were retrieved and qualified for the analysis criteria defined in the manuscript. Authors found that overall proportion of SAE or non-serious AE  of AADs monotherapy in pediatric patients with solid tumors were 46% and 89%, while the proportion was increased in combination therapy. It is a well organized and straightforward manuscript that discusses overall safety profile in a specific population with specific drug class. Overall, the manuscript is clear and relevant to the safely use of AADs in pediatric cancer patients. I do have some requests for revisions: 

1) Page 1 line 17 and 33, please correct the typo "anti-antiangiogenic" to "anti-angiogenic".  Please also consistently use either "antiangiogenic" (title use this version) or "anti-angiogenic" throughout the manuscript.  Also if we have defined antiangiogenic drugs as abbreviations of AADs, please use abbreviations thereafter.

2) In the abstract, please briefly mention the conclusions of AEs in combination therapy as the final sentence.

3) Please put all reference numbers before the period instead of after.    E.g., [1].  instead of  .[1]

4) Page 5 line 174, please provide reference(s) for the random intercept logistic regression model method in R.  

5) Page 5 line 192 said seven additional studies while line 194 showed only 6 clinical trials. Please clarify.

6) Page 3 Figure 3 (**: here page number restarted in those pages without line numbers after Figure 1):  please remove the heterogeneity equations as those were illegible. And please label the column 3 and 4, namely the meaning of those two numbers.   Same comments for Figure 6.

7)  Page 5 and 7, Figure 4 and 5:  Please use higher resolution or make it legible.  Please consider all x-axis label "Incidence (IC 95%)" to be changed into "Median incidence (95% CI)".  Same comments for the supplementary figures.

8)  Page 11 second last paragraph: please correct typo "hypotyroidism" to "hypothyroidism".

9)  Page 13 second paragraph: please keep the first "however", and avoid using "however" multiple times for the transition. Remove or use alternatives. 

Author Response

Reviewer 1

Andrea Spini and colleagues performed a systematic review and meta-analysis on the safety of antiangiogenic drugs either as monotherapy or in combinations in pediatric patients with solid tumors.  Total of 70 studies were retrieved and qualified for the analysis criteria defined in the manuscript. Authors found that overall proportion of SAE or non-serious AE of AADs monotherapy in pediatric patients with solid tumors were 46% and 89%, while the proportion was increased in combination therapy. It is a well organized and straightforward manuscript that discusses overall safety profile in a specific population with specific drug class. Overall, the manuscript is clear and relevant to the safely use of AADs in pediatric cancer patients. I do have some requests for revisions: 

Dear reviewer, thank you for your valuable comments that help us to improve our manuscript.

  • Page 1 line 17 and 33, please correct the typo "anti-antiangiogenic" to "anti-angiogenic".  Please also consistently use either "antiangiogenic" (title use this version) or "anti-angiogenic" throughout the manuscript.  Also if we have defined antiangiogenic drugs as abbreviations of AADs, please use abbreviations thereafter.

Manuscript was modified accordingly.

  • In the abstract, please briefly mention the conclusions of AEs in combination therapy as the final sentence.

Manuscript was modified accordingly.

  • Please put all reference numbers before the period instead of after.    E.g., [1].  instead of  .[1]

Manuscript was modified accordingly.

  • Page 5 line 174, please provide reference(s) for the random intercept logistic regression model method in R.  

Reference was added to the manuscript.

  • Page 5 line 192 said seven additional studies while line 194 showed only 6 clinical trials. Please clarify.

Thank you for this observation. We didn’t notice this oversight.

  • Page 3 Figure 3 (**: here page number restarted in those pages without line numbers after Figure 1):  please remove the heterogeneity equations as those were illegible. And please label the column 3 and 4, namely the meaning of those two numbers.   Same comments for Figure 6.

We believe that the heterogeneity gives precious information to the readers thus we increased the font. We also add the labels of column 3 e 4 in the figure caption. To facilitate the review of the manuscript we also restore the page numbers and the line numbers. 

  • Page 5 and 7, Figure 4 and 5:  Please use higher resolution or make it legible.  Please consider all x-axis label "Incidence (IC 95%)" to be changed into "Median incidence (95% CI)".  Same comments for the supplementary figures.

We enhanced the resolution of the figures. As for the x label, we found that we cannot properly talk of incidence but rather proportion. Figures were changed accordingly.

  • Page 11 second last paragraph: please correct typo " hypotyroidism " to "hypothyroidism".

Manuscript was modified according to reviewer suggestion.

  • Page 13 second paragraph: please keep the first "however", and avoid using "however" multiple times for the transition. Remove or use alternatives. 

Manuscript was modified according to reviewer suggestion.

Reviewer 2 Report

1) Introduction needs more explanation particularly focus on problem and their solution.

2) Why the study was conducted and what is the gap of research and how to find the solution to fill the gap is not clearly defined. 

3) Result portion need more clearity. Some tables are very large and out of paper order.

Author Response

Reviewer 2

  • Introduction needs more explanation particularly focus on problem and their solution.

Dear reviewer, thank you for your valuable comments that help us to improve our manuscript. According to you suggestion introduction section was modified as follow (page 3 line 77-89):

“Although for anti-angiogenic drugs there are no clear predictive biomarkers of response, these drugs are considered promising chemosensitizers of anticancer strategies such as chemotherapy, targeted therapies, and immune therapies in several advanced tumors, and while not approved, they are frequently used in pediatric population [8,9]. For example, benefits of anti-angiogenic therapy in brain tumors are not clear but it has successfully introduced to treat radiation-induced necrosis in several solid tumors [7,8]. To date, there is limited clinical evidence focusing on safety of anti-angiogenic drugs in pediatric patients. This is mainly due to the low prevalence and the heterogeneity of pediatric cancers.”

  • Why the study was conducted and what is the gap of research and how to find the solution to fill the gap is not clearly defined.

As suggested by the reviewer this sentence was modified and added to the introduction section (page 3 line 90-94):

“To date, there is limited clinical evidence focusing on safety of anti-angiogenic drugs in pediatric patients. This is mainly due to the low prevalence and the heterogeneity of pediatric cancers. The purpose of this systematic review was to estimate the proportion of adverse events of anti-angiogenic drugs used to treat solid tumors in patients aged 0-18 years and to assess the potential knowledge gaps on safety of these drugs.”

We also provided some recommendation at the end of the discussion section:

“According to results of this systematic review and meta-analysis we can also provide some recommendations. First, we did not get any information on the long term toxicities and observational studies with long term follow up using routinely collected electronical healthcare data are required. Second, this study did not aim to collect evidence from studies using spontaneous reporting system, however the use of these platforms could be fundamental to fill the gap of the safety of anti-angiogenic drugs in special populations.”

  • Result portion need more clearity. Some tables are very large and out of paper order.

We check the order of tables and figures in the manuscript. Moreover, we fitted table 2 in two pages and we reduced the spacing between the columns. A table with included studies is always reported in systematic reviews. Given the high number of included studies it could not be possible to fit the table in a single page. Finally figure resolution has been enhanced.